# An Insight into the Novel Immunotherapy and Targeted Therapeutic Strategies for Hepatocellular Carcinoma and Cholangiocarcinoma

**DOI:** 10.3390/life12050665

**Published:** 2022-04-30

**Authors:** Eleni-Myrto Trifylli, Evangelos Koustas, Nikolaos Papadopoulos, Panagiotis Sarantis, Georgios Aloizos, Christos Damaskos, Nikolaos Garmpis, Anna Garmpi, Michalis V. Karamouzis

**Affiliations:** 11st Department of Internal Medicine, 417 Army Share Fund Hospital, 11521 Athens, Greece; vang.koustas@gmail.com (E.K.); npnck7@yahoo.com (N.P.); aloizosgio@yahoo.gr (G.A.); 2Division of Molecular Oncology, Department of Biological Chemistry, Medical School, National and Kapodistrian University of Athens, 11527 Athens, Greece; panayotissarantis@gmail.com (P.S.); mkaramouz@med.uoa.gr (M.V.K.); 3N.S. Christeas Laboratory of Experimental Surgery and Surgical Research, Medical School, National and Kapodistrian University of Athens, 11527 Athens, Greece; x_damaskos@yahoo.gr; 4Renal Transplantation Unit, Laiko General Hospital, 11527 Athens, Greece; 5Second Department of Propedeutic Surgery, Laiko General Hospital, Medical School, National and Kapodistrian University of Athens, 11572 Athens, Greece; nikosg22@hotmail.com; 6First Department of Pathology, Medical School, National and Kapodistrian University of Athens, 11527 Athens, Greece; annagar@windowslive.com

**Keywords:** cholangiocarcinoma, hepatocellularcarcinoma, immunotherapy, immune checkpoint inhibitors, immunoresistance, tumor microenvironment

## Abstract

Hepatocellular carcinoma (HCC) and cholangiocarcinoma (CCA) constitute highly malignant forms of primary liver cancers. Hepatocellular and bile duct carcinogenesis is a multiplex process, caused by various genetic and epigenetic alterations, the influence of environmental factors, as well as the implication of the gut microbiome, which was undervalued in the previous years. The molecular and immunological analysis of the above malignancies, as well as the identification of the crucial role of intestinal microbiota for hepatic and biliary pathogenesis, opened the horizon for novel therapeutic strategies, such as immunotherapy, and enhanced the overall survival of cancer patients. Some of the immunotherapy strategies that are either clinically applied or under pre-clinical studies include monoclonal antibodies, immune checkpoint blockade, cancer vaccines, as well as the utilization of oncolytic viral vectors and Chimeric antigen, receptor-engineered T (CAR-T) cell therapy. In this current review, we will shed light on the recent therapeutic modalities for the above primary liver cancers, as well as on the methods for the enhancement and optimization of anti-tumor immunity.

## 1. Introduction

Hepatocellular carcinoma (HCC) and cholangiocarcinoma (CCA) constitute highly malignant forms of primary liver cancers, with the former being the third most frequent cause of death, due to cancer and the main cause of morbidity in cirrhotic patients [1]. The latter represents a rare form of gastrointestinal cancer (3%), associated with increased morbidity, which is attributed to the delayed time of diagnosis when the disease is already advanced [2]. The most common type of primary liver cancer is considered Hepatocellular carcinoma, followed by intrahepatic cholangiocarcinoma (iCA) [3].

HCC is arisen from a multifactorial process, including deregulated signaling pathways, epigenetic and genetic aberrations, as well as the influence of environmental factors and gut microbiota. The main hepatic diseases that predispose to hepatic carcinogenesis are chronic inflammatory conditions such as non-alcoholic steatohepatitis (NASH), alcoholic hepatitis, cholestatic diseases, chronic infection with hepatitis B and C virus (HBV, HCV), hereditary disorders such as hemochromatosis, as well as autoimmune-induced hepatitis, which lead to liver scarring and cirrhosis development [4]. Due to the delayed diagnostic time of HCC, the majority of the tumors are discovered in late-stage conditions, when they are considered inoperable [5,6].

CCA is another highly malignant cancer, which presents not only gender disparity, with males being 1.5 times more predisposed, but also epidemiological alterations based on the geographical region. In western countries is closely associated with chronic inflammation of the biliary tract, such as primary sclerosing cholangitis (PSC), biliary lithiasis, metabolic diseases, as well as chronic hepatic inflammatory diseases, including viral hepatitis B and C, NASH, mainly for intrahepatic cholangiocarcinogenesis. In Eastern countries, two well-demonstrated risks factor is a parasitic infection with larvae of Opisthorchis viverrini, and Clonorchissinesis via food consumption, as well as exposure to aflatoxins [7,8], which is also associated with HCC development. There is a notable association of anatomical sub-entities including, FGFR gene fusions, mutant IDH1/2, ARID1A, and BRAF genes in intrahepatic CCAs (iCCAs), while for extrahepatic tumors mutations in ELF3, IDH1/2, as well as PRKACA/B and FGFER fusions, while, TP53 and KRAS mutations are also reported for both subtypes [9,10].

Immunotherapy is considered a step forward for the management of a large variety of malignancies, especially in advanced stages. These agents are quite specific, targeting the cancer cells, without exhibiting any unfavorable effect on normal cells. The mechanism of immunotherapeutic modalities is either as suppressors or promoters of immune responses, aiming at antigens that are presented in malignant tumors [11]. For the management of gastrointestinal malignancies, including HCC and CCA, the main immunotherapeutic modalities are cancer vaccines, immune checkpoint inhibitors, as well as T cell therapies. Immune checkpoint blockade includes antibodies against (i) cytotoxic T-lymphocyte associated protein-4 (CTLA-4), (ii) programmed death-ligand 1 (PD-L1), and (iii) programmed cell death protein (PD-1). The aforementioned immunological targets have a crucial role in carcinogenesis as the CTLA-4 suppresses the anti-tumor immune responses, while tumors that express PD1 and PD-L1avoid the physiological apoptotic mechanism and tumor cells are continuously multiplied.

The immune response is multiphasic, including (i) the initial attempt of cancer-cell elimination, followed by the second (ii) balance-step, in which the attempt of elimination is not possible, in relation to the progression of the disease. The last step is considered the (iii) tumor-escape, in which the previously asymptomatic disease becomes symptomatic, regardless of the administrated immunotherapy.

Tumor cells escape from the innate immune response, via the expression of antigens on their surface, so their recognition by T cytotoxic cells becomes unfeasible. These antigens are the aforementioned immune checkpoints: PD-1, PD-L1, as well as CTLA-4, which suppress anti-tumor immunity and apoptosis, leading to unrestricted replication [12].

Immunoresistance occurs also in this therapeutic modality via multiple factors, such as the implication of gut microbiome on the anti-tumor immunity, changes in the targets of immune checkpoint inhibition, T cell dysfunction, and exhaustion, as well as the suppressive effect of various immune cells on the tumor microenvironment (TME) [13,14,15]. Some of these immunosuppressant cells are B and T regulatory cells, myeloid-derived suppressor cells (MDSC), as well as tumor-associated macrophages TAMs, and Cancer-associated fibroblasts CAFs [15].

Conventional chemotherapeutic agents are considered ineffectual for HCC [16], however immunotherapeutic strategies including the use of multi-kinase inhibitors, monoclonal antibodies, as well as cancer vaccines, T cell, and oncolytic virus therapies, are some of the treatment modalities that present favorable effects and enhancement of overall survival for patients with advanced cancer [17,18]. Gut microbiome modulation presents another potent method of intensification of anti-tumor immunity, which is implied by numerous reports that demonstrate the close interrelation of intestinal microbiota dysbiosis and liver pathogenetic mechanisms, including carcinogenesis [19]. Gemcitabine/cisplatin combination therapy is considered the first-line treatment for advanced cholangiocarcinoma. However, the effectiveness of the immunotherapeutic strategy for CCA is still under study. The modulation of the tissue TME in CCA has a crucial role for cholangiocarcinogenesis, as it inhibits the anti-tumor immunity, via a complex process, the so-called T cell exclusion mechanism [20]. Checkpoint immune blockade, as well as adaptive T cell treatment and cancer vaccines, are also applied in CCA, however, vaccines proved ineffective [21]. It is worth to be mentioned that CCA constitutes a malignancy with many druggable targets. However, targeted therapeutic strategies in CCA are generally doubted in clinical practice, due to the fact that it does not significantly improve survival [22,23].

In this current review, we will shed light on the recent immunotherapeutic and targeted modalities for the management of HCC and CCA, as well as on the methods for the enhancement of anti-tumor immunity.

## 2. Immunotherapy Strategies for HCC

Immunotherapy is a novel therapeutic strategy that has a crucial role in the management of gastrointestinal cancers, including hepatobiliary malignancy, widely used, especially in advanced stages, when the tumors are already inoperable at the time of diagnosis.

### 2.1. Multi-Kinase Inhibitors for HCC

Two of the well-studied oral multi-kinase inhibitors that constitute first-line treatment for HCC are sorafenib and lenvatinib. The former agent demonstrated a moderate enhancement of the overall survival versus placebo, based on SHARP [24] and ORIENTAL [25] phase III clinical trials, while the latter in the non-inferiority study (REFLECT), proved non-inferior to sorafenib, with 13.6 months mean survival [18]. Both of the agents aim at multiple targets, such as sorafenib, which targets the receptor of platelet-derived growth factor (PDGF-R), c-Kit, vascular endothelial growth factor (VEGFR), as well as Raf and Flt3. Lenvatinib aims at similar targets, including FGFR and RET. Second-line therapeutic agents are Cabozantinib and Regorafenib, which have been evaluated in phase III clinical studies versus placebo, the CELESTIAL [26] and RESORCE [27], respectively. Cabozantinib demonstrated favorable improvement of survival, while Regorafenib proved beneficial as well, despite the former dose of sorafenib, without the aggravation of the side effects. There are multiple other pharmaceutical molecules that are still in the phase II studies and showed favorable results for HCC management, such as axitinib, a tyrosine kinase inhibitor (TKI), combined with transarterial chemoembolization (TACE), which demonstrated notable effectiveness for advanced HCC [28], decitabine, which was found beneficial in low doses, without remarkable toxicity [29] and nintedanib that showed effects alike sorafenib [30]. The molecular targets of the above-mentioned drugs and the biochemical pathways for their effect are shown in Figure 1**.**

### 2.2. Immune Checkpoint Inhibition for HCC

Based on phase II study, Bevacizumab that targets VEGF, showed favorable effects on neoangiogenesis [31], while based on phase I trial combinational treatment with bevacizumab and atezolizumab, a PD-L1 inhibitor, proved beneficial for overall survival and increased free-of disease period in comparison with monotherapy with sorafenib [31]. Another phase III clinical study, CheckMate-459 made a comparison between nivolumab, a PD-1 inhibitor, and sorafenib, however, it did not show clear superiority of the single-use of nivolumab, for which further research is needed [32]. Pembrolizumab is another PD-1 inhibitor that is studied in the phase II trial KEYNOTE-224 and presents a notable anti-neoplastic effect on advanced HCC cases [33]. The mechanism of immune checkpoint inhibition is demonstrated in Figure 2.

We demonstrate in Table 1 some of the main immunotherapy agents that are either utilized in clinical practice or still studied in phase I/II.

### 2.3. Cell-Based Immunotherapy for HCC

There are various cell-based immunotherapies for HCC, including CAR-Tcell therapy, mesenchymal stem cells (MSCs) therapy, with the utilization of either bone or adipose mesenchymal stem cells [34], as well as Cytokine-induced killer cells (CIKs) [35]. It is reported that overall survival is significantly improved in the case of CIK therapy, which has a direct cytotoxic anti-tumor effect, while CAR-T cell treatment limits the tumor escape mechanism. Tumor escape mechanism prevention is crucial for the management of immune-resistant HCC (Figure 2).

### 2.4. Oncolytic Viral Therapy for HCC

Herpes simplex virus 1 (HSV-1), is considered ideal as a vector for virotherapy [36,37], the so-called oncolytic ICP0-null virus (d0-GFP), which is either intravenously administrated or intratumorally, which were proven beneficial for the patients with a variety of malignancies, including HCC [37]. Another viral therapy includes an armed oncolytic virus, such as vaccinia virus, which induces an anti-tumor immune response by releasing cytokine signals, against HCC cell lines [38].

### 2.5. Cancer Vaccines for HCC

Some of the vaccine-based immunotherapy strategies include tumor-specific neoantigens vaccine, based on the so-called HEPAVAC project [39,40], as well as, peptide vaccines, like the glypican-3 (GPC3) vaccine, which is based on the overexpression of GPC3 on the tumor cell membrane in HCC. Based on the phase I study results, HCC patients that received this vaccine, expressed GPC3-specific cytotoxic T cells [41,42]. Another type of vaccine is the dendritic cells (DC) vaccine, which induces the elimination of tumor cells [43,44]. Moreover, there are studies about the combinational treatment of DC vaccine with nivolumab [45], as well as with PD-L1 inhibitor, with the latter combination having a beneficial effect on HCC [46].

### 2.6. Gut Microbiome Modulation for Optimal Anti-Tumor Response in HCC

The importance of the gut microbiome was relatively neglected in the last decades, however, the recent years there are multiple studies about the significance of the above entity in immune responses, signaling, and metabolic pathways, as well as carcinogenesis, including HCC. There is an interrelation of gut and liver functions, the so-called gut-liver axis, which is achieved via the hepatic portal circulatory system. Microbial products, resulting from microbial dysbiosis, closely influence the liver function and derivatives, such as the bile acids (BAs), which further leads to hepatobiliary carcinogenesis [47]. Microbial dysbiosis, lead to the disruption of the gut barrier, which leads to the release of microbial metabolites in the portal circulation, influences the BAs synthesis and construction, resulting in a variety of modified BAs, which are closely associated with a wide spectrum of malignant tumors in the gastrointestinal tract [48]. The altered BAs have a crucial impact on FXR and TGR5 receptors of the host, leading to an inflammatory reaction, deregulation of various signaling pathways, deregulated multiplication of cells, leading to tumor development and carcinogenesis [49], cirrhosis is significantly associated with gut microbiota dysbiosis, which, is highly connected with immunoresistance [50].

Modulation of gut microbiota constitutes a potent method of manipulation of the anti-tumor immunity [51]. Based on a recent study of patients with immunoresistance to sorafenib and recurrence of HCC, and their anti-tumor response to ICIs (anti-PD-1) treatment, patients that presented no response to treatment, had in their microbiome overproduction of Proteobacteria, in comparison to those with the optimal response, which had an increased amount of *Ruminococcus* spp. and *Akkermansia muciniphila* in their fecal samples [52]. The administration of antibiotics, such as Vancomycin with immunotherapy modalities, such as ICIs, could modify the gut microbiota and limit microbial dysbiosis [51]. Moreover, another possible modulation of the gut-liver axis could be FXR agonists for the optimization of immunotherapy response and the regulation of BAs function [49]. The utilization of next-generation probiotic supplements, which include favorable, beneficial species of microbiota, such as *Clostridium XIVa* and *IV*, *Akkermansia muciniphila,* which suppress the overproduction of those, who induce dysbiosis, can lead to a reduction of the risk for HCC development. Meanwhile, an alteration of the viral load of hepatitis virus B and C and reduction of the hepatocellular damage that they cause HCC can also be achieved via the administration of probiotics, including, *E. faecalis* [53]. Intake of Bifidobacterium, can also improve the response to ICIs against PD-L1 target and modify the TME for HCC, as well as in melanoma [54], while intake of Firmicutes and Faecalibacterium supplements can enhance the response of monoclonal antibodies against CTLA-4, such as ipilimumab, as it was reported also in melanoma cases. Fecal microbiota transplantation in combination with oral microbiota supplementation with *Akkermansia muciniphila* can alter immunoresistance to PD-1 inhibition, as it was reported in mice tumors, by alteration of TME anti-tumor response and immune cells recruitment [55].

## 3. Immunotherapy Strategies for CCA

### 3.1. Small-Molecule Kinase Inhibitors (SMKIs) for CCA

Based on the molecular landscape of CCA many fusions of FGFR, are reported. FGFR inhibition constitutes a potent anti-cancer therapeutic modality that is currently under preclinical trials, especially for iCCA [56]. In phase II study of infigratinib BGJ398 in patients with late-stage CCA, chemoresistant to platinum anti-neoplastic drugs, proved significantly beneficial [57]. The use of another SMKI is ARQ 087 is currently in phase I/II for iCCA and other solid malignant tumors, which showed also efficiency [58]. Moreover, there are also non-selective SMKIS, such as pazopanib [59] and ponatinib [60] for end-stage iCCA cases, including genetic aberrations such as gene fusions of FGFR2-MGEA5. The use of single-therapy ponatinib proved remarkably beneficial for the limitation of tumor progression, dissemination of cancer cells, limitation of lymphadenopathy, based on phase II studies. Based on the FOENIX-CCA2 trial, the use of irreversible pa-FGFR1-4 inhibitor, futibatinib, has remarkable benefits for iCCA, presenting a fusion of FGFR [61]. The utilization of AZD457 and BGJ398 inhibition, have proved also efficient for tumors that exhibit resistance [56,62]. Furthermore, there is also reported the combinational therapy of SMKIs with MEK blockers, such as pazopanib and trametinib, respectively, as well as the use of another MEK blocker, selumetinibis also studied for CCA [63,64]. Additionally, there is a phase I clinical trial about the utilization of MET kinase inhibitors, such as tivantinib [65] for advanced CCA, or combined with a cytotoxic chemotherapeutic agent, gemcitabine with relative beneficial effect [66]. Another novel mutation in CCA is IDH1/2, in which selective inhibition, such as with AGI-6780 for iCCA, provided favorable results with good toleration [67].

### 3.2. Immune Checkpoint Blockade in CCA

Based on phase II study of pembrolizumab, a PD-1 inhibitor with a combination of the chemotherapeutic scheme of mFOLFOX6 (leucovorin, 5-fluorouracil, oxaliplatin) for advanced CCA [68], however, the application of pembrolizumab for CCA is still under study. As a single-agent treatment, pembrolizumab does not present a remarkable efficiency for anti-tumor management, with a response of 5.8% and a poor improvement of overall survival (7.4 months) [69]. Another PD-1 inhibitor, nivolumab, was also studied as a single-agent treatment for CCA, with more favorable results than the aforementioned agent [70]. There is an ongoing phase II study about the combinational treatment with durvalumab (monoclonal antibody against PD-L1), paclitaxel, and tremelimumab (CTLA-4 inhibitor) in CCA [71]. Inhibition of PD-L1, via the use of durvalumab, is studied in phase I/II clinical trial, however single use of durvalumab for CCA is not studied yet, in comparison with its utilization as monotherapy in HCC, which presented a moderate improvement for HCC, compared to other ICIs. There is a relatively recent PD-1 and TGF-b inhibitor, bintrafusp alfa or M7824, which proved efficient for immune-resistant CCA, while there is a clinical trial in phase I that is currently performed (NCT02699515), for cases of recurrent CCA after the utilization of first-line chemotherapeutic agents, with 12.7 months, median overall survival [72,73]. Some of the immunotherapeutic drugs are demonstrated in Table 2.

### 3.3. Cancer Vaccines in CCA

As in HCC cases, also in CCA vaccines development are also reported, such as the DC and peptide vaccines, however without presenting adequately favorable anti-neoplastic results [74], mainly attributed to T cell exclusion mechanism and TME-associated immunosuppression, by which TME prevent the aggregation of T cells that can induce harmful effects to tumors. T cell exclusion constitutes a difficult task to be faced for the achievement of the optimal response to immunotherapy [75].

### 3.4. CAR-T Cell Therapy for CCA

The concept of recombinant T cells with specificity for killing the cancer cells, constitutes a possible potent therapeutic modality, such as the use of anti-CD133 CAR T-cells (4th generation) for CCA that express CD133, provided beneficial effect on the tumors, however, due to its possible toxicity, the use of this modality is still under study. There are also studies about the CAR-T cells against CD133, as well as epidermal growth factor receptor (EGFR), with also favorable anti-tumor effect [13].

### 3.5. The Implication of Gut-Microbiome in CCA

As it was referred to above, in western countries CCA development is closely associated with chronic inflammation of the biliary tract, such as PSC, biliary lithiasis, metabolic diseases, as well as chronic hepatic inflammatory diseases, including HCV, HBV, NASH, mainly for intrahepatic cholangiocarcinogenesis. The gut microbiome can alter the function of BAs and lead to the formation of gallbladder lithiasis, which further modifies the intestinal microbiome and leads to dysbiosis, resulting in the overgrowth of Oscillospira, and Proteobacteria, while other microbes such as Roseburia were significantly reduced. Meanwhile, in cases of PSC, an overgrowth of Veillonella was reported [76]. Focusing on fecal samples in CCA patients, the amount of Lactobacillus, Alloscardovia, Actinomyces, as well as Peptostreptococcaceae was notably increased, whiles the amount of Leuconostocaceae and Ruminococcus, and was decreased, which implies their role in cholangiocarcinogenesis. Meanwhile in blood cultures were found Actinobacteria, Proteobacteria, Firmicutes, cyanobacteria, and Bacteroidetes significantly increased, while in bile specimens some of the bacteria found overgrown were, *E. coli*, Enterococcus faecium, and faecalis, Nitrospirae, as well as Enterobacter cloacae. The microbiomecan possibly beused as a druggable target for CCA therapy, as well as FMT that can help with the condition of dysbiosis and the optimization of anti-tumor immunity, however, this therapeutic modality needs further research [76,77,78].

### 3.6. Mechanisms for Overcoming Immunoresistance

TME constitutes the principle mechanism of immunoresistance, which can be overcome by multiple therapeutic strategies, such as the utilization of (i) angiogenesis inhibitors, that could possibly promote the anti-tumor response to ICIs, while an ongoing study in phase I/II, demonstrates anti-PD1 agent, pembrolizumab, combined with VEGFR inhibitor, lenvatinib for end-stage HCC [79].

One of the strategies is the (ii) inhibition of the CXCL12/CXCR4 pathway, as well as (iii) MDSC inhibition, achieved by PI3K inactivators or combinational therapy composed of (iv) radiotherapy and ICIs, including PD-1/PD-L1inhibitors. Epigenetic aberrations such as histone modifications and DNA hypermethylation, inhibit the secretion of chemokines CXCL9 and CXCL10 by T-helper 1 cell (Th1-cell). The above phenomenon could be limited via the utilization of (v) epigenetic modulatory agents, which not only stimulate the T cell infiltration inside the tumor but also lead to the enhancement of the response to PD-L1inhibitors. Moreover, another mechanism for immunoresistance management is (vi) Chemokine-targeted therapies, including the increase of the regulation for CXCL11,10, and 9. Another method is considered the (vii) activation of toll-like receptors (TLRs), as well as the utilization of heparin-based anticoagulant treatment [80].

## 4. Adverse Effects of Immunotherapeutic and Targeted Agents

Despite the major favorable effects of all the aforementioned agents, all of them demonstrated a different level of toxicity. There are many manifestations of immune-related toxicity, such as on the skin with the most common symptoms being pruritus, as well as rash and vitiligo, while xerosis cutis, alopecia areata, and stomatitis are some other rare immune-related skin AEs that are demonstrated after the use of immune checkpoint blockade. Life-threatening manifestations that require immediate hospitalization, have been reported after the utilization of checkpoint inhibitors, such as Steven-Johnson syndrome, toxic epidermal necrolysis (TEN), DRESS syndrome, as well as Sweet syndrome, presented by acute febrile neutrophilic dermatosis. Gastrointestinal (GI) toxicity constitutes a common, well-studied entity of AE, especially after the use of anti-CTLA4 agents. Almost half of the patients (27% up to 54%) presented diarrhea with or without colitis, while cases of enterocolitis that were attributed to the utilization of anti-CTLA4 agent, were also closely associated with concomitant use of non-steroidal anti-inflammatory drugs. Moreover, these patients who present enterocolitis, have a distorted endoscopic examination of the upper GI system, presenting patchy lesions, macroscopically and modifications in the crypt-villus units, as well as infiltration of inflammatory cells, microscopically. Additionally, another type of GI AE is acute colitis, which is reported after the use of anti-PD-1 agents, with manifestations also from the upper GI tractas well as acute colonic pseudo-obstruction. Hepatitis is also reported as an AE of ICIs, usually resolved in 4–6 weeks, while it must be excluded a recurrent CMV infection, hepatotoxicity, due to other medication, as well as other viral hepatitis. Endocrinological manifestations can also occur, such as thyroid deregulation, commonly reported after the use of ICIs. Hypophysitis constitutes a severe AE, mainly reported after anti-CTLA4 treatment, which was attributed due to the development of antibodies against the pituitary gland, the induction of complement cascade, as well as the infiltration of the gland with mononuclear cells, whereas de novo diabetes is also reported mainly after the use of anti-PD-1/anti-PD-L1 agents. Respiratory manifestations are also reported, such as immune-related pneumonitis after the use of anti-PD1/anti-PD-L1 agents, with the most severe AE being the DADsyndrome, including diffuse damage of the alveoli, a hallmark of acute respiratory distress syndrome (ARDS). Last but not least, cardiac (<1%), renal (<1%), and neural toxicity (1%) are also reported, however, they constitute very rare AEs [81].

For targeted therapeutic agents, it is reported that patients that received sorafenib versus lenvatinib presented similar AEs such as diarrhea (46% vs. 39%), hypertension (30% vs. 42%), and anorexia (27% vs. 34%), while for sorafenib was also reported animmune-related skin manifestation, such as palmar-plantar erythrodysesthesia (52%) [17,18], while for regorafenib, the most common AEs were hypertension, fatigue, as well as hand-foot syndrome [27]. Whereas, for SMKIs it is reported cardiotoxicity with a wide variety of manifestations such as arrhythmia, decreased left ventricular ejection fraction, cases of myocardial infarction, and even heart failure [82]. Finally, hepatic [83] and skin toxicity [84] are also reported for a wide variety of SMKIs.

## 5. Conclusions

The therapeutic management of HCC and CCA is considered a difficult task for clinicians, due to their multiplex molecular landscapes including various genetic and epigenetic modifications in combination with the impact of gut microbiota and environmental factors. Novel immunotherapeutic agents open the horizon for new treatment strategies, aimed at increased overall survival, especially for chemoresistant malignant tumors. New modalities of immunotherapy such as CAR-T cell treatment, cancer vaccines, oncolytic viral treatment, gut microbiome modulation and targeted therapy are some extra “weapons” against these highly aggressive malignancies. The development of novel ICIs, proved relatively beneficial for the improvement of anti-tumor response and overall survival, while modification of TME, prevention of T cell exclusion, as well as the tumor escape mechanism, constitute some of the methods for the management of immunoresistance. The gut microbiome was relatively underrated for its significance in many metabolic and signaling pathways, as well as in carcinogenesis. Nowadays, it is in the spotlight, while it is considered an important method of immunotherapy response enhancement. Finally, immunotherapy is remarkably important for cancer management, especially due to its selective anti-neoplastic effect on cancer cells, being harmless for healthy tissues. However, further research is needed for the discovery of more potent agents that significantly can improve the survival of cancer patients, while further study is also needed for the development of other possible methods that can overcome the immunoresistant malignant tumors.

## Figures and Tables

**Figure 1 life-12-00665-f001:**
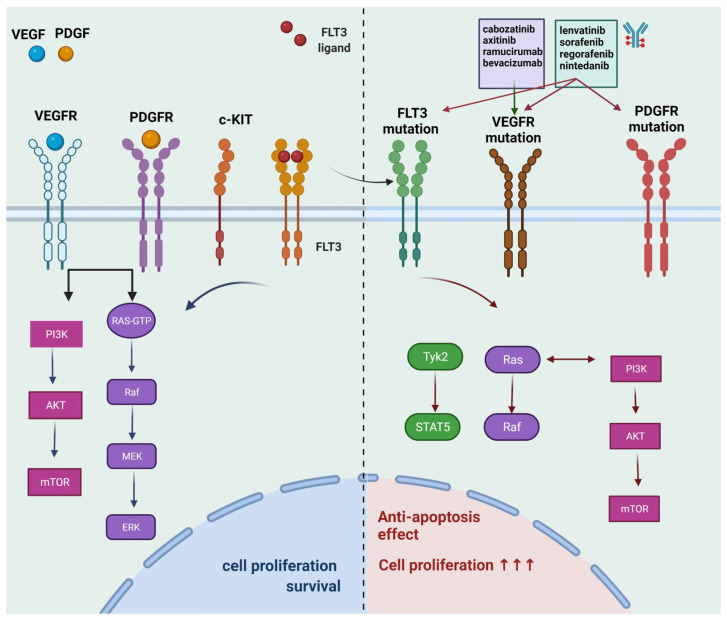
**A schematic presentation of the molecular targets of multikinase and tyrosine kinase inhibitors.** Some of the pathways that the VEGF signaling cascade includes are: the Ras/MAPK and PI3K/AKT pathway. The former is implicated in gene expression, as well as cell proliferation, while the latter in cell survival. Similarly, the PDGF and FLT3 signaling cascade are also related to PI3K/AKT and Ras/MAPK pathways, with the latter being closely associated with cell proliferation, differentiation, and survival. The mutation of all the above lead to over-proliferation and enhanced survival. In this scheme we demonstrate some of the main inhibitors for FLT3, VEGFR and PDGR mutant receptors. The figure was created with BioRender.com (accessed on 27 April 2020) (QN23UJU88O agreement number).

**Figure 2 life-12-00665-f002:**
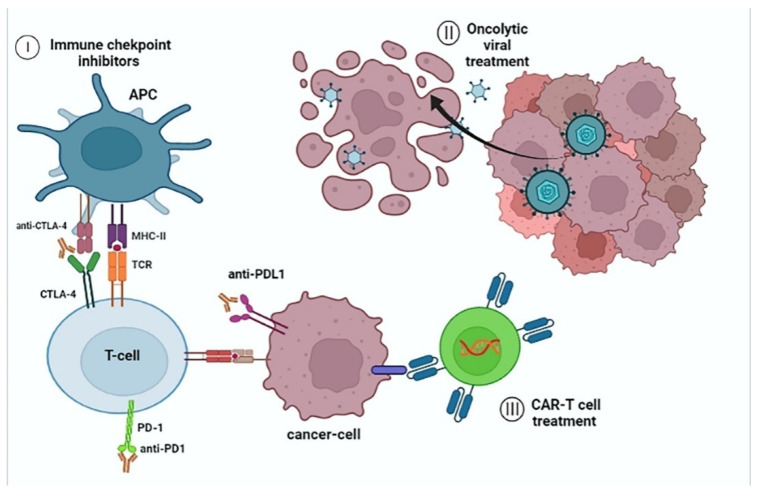
**Schematic drawing demonstrating some of the main immunotherapeutic modalities.** Tumor cells escape from the innate immune response, via the expression of antigens on their surface, such as the immune checkpoints: PD-1, PD-L1, as well as CTLA-4, so their recognition by T cytotoxic cells becomes unfeasible, resulting in the suppression of the anti-tumor immunity and apoptosis, leading to unrestricted replication. (**I**) Immune checkpoint inhibitors block those proteins, allowing the T-cells to destroy the cancer cells. Another therapeutic modality is (**II**) oncolytic viral treatment, via the intravenous or intratumoral administration of an oncolytic virus, such as Herpes simplex virus 1 (HSV-1) or vaccinia virus, which induces an anti-tumor immune response by releasing cytokine signals, against the cancer cell lines. (**III**) CAR-T cell treatment includes the genetic manipulation of T-cells and the construction of recombinant T-cell receptors for the destruction and elimination of malignant cells. Figure was created with BioRender.com (accessed on 27 April 2020) (LG23TTM6SH agreement number). APC (Antigen-presenting cells); MHCI/II (major histocompatibility complex I/II); (TCR) T-cell receptor; (CAR) Chimeric antigen receptor; (CTL4) cytotoxic T-lymphocyte-associated antigen 4; (PD-1) Programmed Cell Death Protein 1; (PD-L1) Programmed death-ligand 1.

**Table 1 life-12-00665-t001:** Immunotherapeutic drugs in HCC.

Pharmaceutical Agent	Molecular Target	Phase of Clinical Trial
Immune checkpoint inhibitors		
Pembrolizumab	PD-1	phase III trial [33]
Nivolumab	PD-1	phase III trial [32]
Atezolizumab	PD-L1	phase III trial [31]
Multikinase inhibitors		
Lenvatinib	PDGF-R, PDGF-R, c-Kit, VEGFR, Raf, Flt3	phase III trial [18]
Sorafenib	RET, PDGF-R, c-Kit, VEGFR, Raf, Flt3	phase III trial [24,25]
Regorafenib	RET, PDGF-R, KIT, VEGFR, RAF, Flt3, TIE2	phase III trial [27]
Cabozantinib	AXL, VEGFR, MET	phase III trial [26]
Nintedanib	RET, PDGF-R, c-Kit, VEGFR, Raf, Flt3	phase II trial [30]
Decitabine	DNA methylationphase	phase trial I/II [29]
Tyrosine kinase inhibitors		
Axitinib	VEGFR	phase II trial [28]
Various monoclonal antibodies		
Ramucirumab	VEGFR	phase III trial [D136]
Bevacizumab	VEGFR	phase II trial [31]

**Table 2 life-12-00665-t002:** Immunotherapeutic drugs in CCA.

Pharmaceutical Agent	Molecular Target	Phase of Clinical Trial
Immune checkpoint inhibitors		
Pembrolizumab	PD-1	phase II trial [69]
Nivolumab	PD-1	phase II trial [70]
Durvalumab	PD-L1	phase I/II trial [71]
Tremelimumab	CTLA-4	phase II trial [71]
Bintrafusp alfa	PD-1, TGF-b	phase I trial [73]
Small-molecule kinase inhibitors		
Infigratinib	FGFR	phase II trial [57]
Pazopanib	PDGF, VEGFR, c-kit	phase I trial [59]
Ponatinib	PDGF, VEGFR2, Scr, FGFR1	phase II trial [60]
Futibatinib	FGFR	phase III trial [61]
Tivantinib	MET	phase I trial [65]
Trametinib	MEK1/2	phase II trial [64]
Selumetinib	MEK1/22	phase II trial [63]

## Data Availability

Data sharing not applicable. No new data were created or analyzed in this study. Data sharing is not applicable to this article.

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
