# Peer review of "An Insight into the Novel Immunotherapy and Targeted Therapeutic Strategies for Hepatocellular Carcinoma and Cholangiocarcinoma"

_life, 2022, doi:10.3390/life12050665_

Round 1

Reviewer 1 Report

An interesting review on modern methods of targeted and immunotherapy for hepatocellular and cholangicellular carcinoma. The review provides current strategies for the treatment of these diseases, based on the use different therapeutic strategies and drugs (tyrosine kinase inhibitors, Cell-based immunotherapy, Oncolytic viral therapy, Cancer vaccines, etc.).  Attention is paid to such a factor of carcinogenesis  as the intestinal microbiota. The mechanisms for overcoming immunoresistance are described. In my opinion, the review can be published, but there are some comments and suggestions for the authors.

  1. The term "immunotherapy strategies" is used in the title. However, for example, the use of tyrosine kinase inhibitors cannot be called immunotherapy, since their effect is not mediated by immune mechanisms. The same can be said about oncolytic viruses and correctors of the intestinal microbiota. Therefore, I propose to add the term “immune and target therapy” to the title.
  2. The authors quite widely cover the results of clinical studies of modern tyrosine kinase inhibitors and monoclonal antibodies, but pay little attention to the molecular mechanisms of their action. In my opinion, it would be advisable to create a drawing similar to Figure 1, in which the molecular targets of the above-mentioned drugs and the biochemical pathways for their effect are shown.
  3. The article practically does not mention the side effects of the described drugs. However, this aspect of their application is very important. I think that more serious attention should be paid to this, it is possible to create a special subsection dedicated to side and undesirable effects.
  4. Technical note. Subsection 2.2. (line 133) is distinguished on the basis of the phase of clinical trials, while other subsections are distinguished on the basis of the pharmacological group of the drugs. This subsection should be combined with subsection 2.1.

Author Response

April 28, 2022

LIFE

RE: Submission of REVISED ‘Mini-Review paper’ (life-1702284)

Dear Editor

Please find enclosed our REVISED Mini-Review entitled “An insight into the novel immunotherapy and targeted therapeutic strategies for hepatocellular carcinoma and cholangiocarcinoma.” to be considered for publication. We would like to thank you and the reviewers for your thoughtful evaluation of our manuscript and your most welcome comments/suggestions. Accordingly, we have now revised our manuscript thoroughly to reflect these comments.

Please find below a point-by-point response to ALL the issues raised by the Reviewers:

Reviewer 1

An interesting review on modern methods of targeted and immunotherapy for hepatocellular and cholangicellular carcinoma. The review provides current strategies for the treatment of these diseases, based on the use different therapeutic strategies and drugs (tyrosine kinase inhibitors, Cell-based immunotherapy, Oncolytic viral therapy, Cancer vaccines, etc.).  Attention is paid to such a factor of carcinogenesis  as the intestinal microbiota. The mechanisms for overcoming immunoresistance are described. In my opinion, the review can be published, but there are some comments and suggestions for the authors.

  1. The term "immunotherapy strategies" is used in the title. However, for example, the use of tyrosine kinase inhibitors cannot be called immunotherapy, since their effect is not mediated by immune mechanisms. The same can be said about oncolytic viruses and correctors of the intestinal microbiota. Therefore, I propose to add the term “immune and target therapy” to the title.

AUTHOR RESPONSE: We thank the reviewer. We have made the appropriate editing. The new title of the manuscript is presented below:

‘‘An insight into the novel immunotherapy and targeted therapeutic strategies for hepatocellular carcinoma and cholangiocarcinoma. ’’

2. The authors quite widely cover the results of clinical studies of modern tyrosine kinase inhibitors and monoclonal antibodies, but pay little attention to the molecular mechanisms of their action. In my opinion, it would be advisable to create a drawing similar to Figure 1, in which the molecular targets of the above-mentioned drugs and the biochemical pathways for their effect are shown.

AUTHOR RESPONSE: We thank the reviewer. We have made the appropriate editing. A new figure (Figure 1), which presents the molecular targets of the targeted therapy and the biochemical pathways for their effect, was added. Accordingly the previously Figure 1 (Schematic drawing demonstrating some of the main immunotherapeutic modalities) now is mentioned as Figure 2.

3. The article practically does not mention the side effects of the described drugs. However, this aspect of their application is very important. I think that more serious attention should be paid to this, it is possible to create a special subsection dedicated to side and undesirable effects.

AUTHOR RESPONSE: We thank the reviewer. We have made the appropriate editing. A new section with the side effects of described drugs is added. The changes in the manuscript are presented below:

           ‘‘4. Adverse effects of immunotherapeutic and targeted agents

 Despite the major favorable effects of all the aforementioned agents, … and skin    toxicity [84] are also reported for a wide variety of SMKIs. ’’

The references list was revised accordingly. Four more references were added (number 81-84).

4. Technical note. Subsection 2.2. (line 133) is distinguished on the basis of the phase of clinical trials, while other subsections are distinguished on the basis of the pharmacological group of the drugs. This subsection should be combined with subsection 2.1.

AUTHOR RESPONSE: We thank the reviewer. We have made the appropriate editing. The subsection 2.2 was merged with 2.1.

The numbering of subsection was also been revised.

Trusting that we have adequately addressed the reviewers' concerns, we would like to thank you for your help in improving our work significantly.

Kind regards,

Eleni-Myrto Trifylli, MD

Reviewer 2 Report

The review entitled: “An insight into the novel immunotherapy strategies for hepatocellular carcinoma and cholangiocarcinoma” by Trifylli et al, is very interesting and informative. The authors provide a valuable review of the immunotherapeutic agents for HCC and cholangiocarcinoma. The tables are very informative. Moreover, the figure is well designed. There are small letter mistakes and some minor grammatical errors. In table 2, the reference number 69 is appeared twice.

Author Response

April 28, 2022

LIFE

RE: Submission of REVISED ‘Mini-Review paper’ (life-1702284)

Dear Editor

Please find enclosed our REVISED Mini-Review entitled “An insight into the novel immunotherapy and targeted therapeutic strategies for hepatocellular carcinoma and cholangiocarcinoma.” to be considered for publication. We would like to thank you and the reviewers for your thoughtful evaluation of our manuscript and your most welcome comments/suggestions. Accordingly, we have now revised our manuscript thoroughly to reflect these comments.

Please find below a point-by-point response to ALL the issues raised by the Reviewers:

Reviewer 2

The review entitled: “An insight into the novel immunotherapy strategies for hepatocellular carcinoma and cholangiocarcinoma” by Trifylli et al, is very interesting and informative. The authors provide a valuable review of the immunotherapeutic agents for HCC and cholangiocarcinoma. The tables are very informative. Moreover, the figure is well designed. There are small letter mistakes and some minor grammatical errors. In table 2, the reference number 69 is appeared twice.

AUTHOR RESPONSE: We thank the reviewer. We have made the appropriate editing. Several grammatical mistakes through out of manuscript were revised. In table 2 the second references 69 was deleted.

Trusting that we have adequately addressed the reviewers' concerns, we would like to thank you for your help in improving our work significantly.

Kind regards,

Eleni-Myrto Trifylli, MD
